# SRSF3 Is a Critical Requirement for Inclusion of Exon 3 of BIS Pre-mRNA

**DOI:** 10.3390/cells9102325

**Published:** 2020-10-19

**Authors:** Ji-Ye Baek, Hye-Hyeon Yun, Soon-Young Jung, Jeehan Lee, Kyunghyun Yoo, Jeong-Hwa Lee

**Affiliations:** 1Department of Biochemistry, The Catholic University of Korea, 222, Banpo-daero, Seocho-gu, Seoul 06591, Korea; baekjiye912@daum.net (J.-Y.B.); nice1205@hanmail.net (H.-H.Y.); syjjeong@hanmail.net (S.-Y.J.); bg597@nate.com (J.L.); ted13579@hanmail.net (K.Y.); 2Institute of Aging and Metabolic Diseases, College of Medicine, The Catholic University of Korea, 222, Banpo-daero, Seocho-gu, Seoul 06591, Korea; 3Department of Biomedicine & Health Sciences, College of Medicine, The Catholic University of Korea, 222, Banpo-daero, Seocho-gu, Seoul 06591, Korea

**Keywords:** BIS, SRSF3, splicing, HSF1

## Abstract

BCL-2 interacting cell death suppressor (BIS), also known as BAG3, is a multifunctional protein. Aberrant expression and mutation of BIS have been implicated in cancers and myopathy. However, there have only been a few studies on the splicing of BIS pre-mRNA. In the present study, through RT-PCR and sequencing in various cell lines and mouse tissues, we identified for the first time the presence of BIS mRNA isomers in which exon 3 or exons 2–3 are skipped. We also demonstrated that the depletion of SRSF3 promoted the skipping of exon 3 of BIS pre-mRNA in endogenous BIS and the GFP-BIS minigene. SRSF3 specifically interacts with the putative binding sites in exon 3, in which deletion promoted the skipping of exon 3 in the GFP-BIS minigene, which was comparable to the effect of SRSF knockdown. Even though acceleration of exon 3 skipping was not observed in response to various stimuli, SRSF3 depletion, accompanied by the production of a truncated BIS protein, inhibited the nuclear translocation of HSF1, which was restored by the wild-type BIS, not by exon 3-depleted BIS. Therefore, our results suggested that the maintenance of SRSF3 levels and subsequent preservation of the intact BIS protein is an important factor in modulating HSF1 localization upon cellular stress.

## 1. Introduction

BCL-2 interacting cell death suppressor (BIS), also known as BAG3, is a 75-kDa protein that has several distinct domains, such as a BAG domain, a WW domain, a proline-rich repeat, and two conserved IPV (Ile-Pro-Val) motifs [1,2]. Through these distinct domains, BIS interacts with different molecular partners, which might underlie the diverse biological function of BIS, including modulation of apoptosis, autophagy, stress response, migration, invasion, as well as cytoskeleton organization [3,4]. The pro-survival function of BIS is linked to its aberrant expression in most human cancers [5,6]. However, in normal tissues, constitutive high expression of BIS is not observed, except in the heart and skeletal muscles [2,6]. Instead, BIS expression is significantly induced upon various stresses, mainly through heat shock factor (HSF) 1 and heat shock response elements (HSE) in the BIS gene promoter [7,8,9]. In addition to HSF1, BIS was also reported to be the target gene of WT-1, contributing to the pro-survival function of WT-1 in leukemic cells [10]. Moreover, FGF-2-driven BIS expression is mediated by Egr-1, while serum deprivation downregulates BIS via the reduction of c-Jun at the transcriptional level [11]. On the other hand, BIS is the downstream target of the endoplasmic reticulum-bound transcription factor AIbZIP/CREB3L4 in prostate cells [12]. The induction of BIS, in turn, affects a variety of cellular events required for cell survival and physiological functions. Thus, these findings indicate BIS expression to be precisely regulated by the appropriate transcription factors, which depends on the cell environment and the cell’s necessity.

While the involvement of various transcriptional factors in BIS expression has been comprehensively studied, there has been little research on the processing of BIS pre-mRNA or post-translational modification of BIS. So far, two types of BIS proteins have been reported: the 75-kDa, full-length gene product and the shorter 40-kDa form, which is detected in neuronal synaptosomes [13]. It has not been determined whether this shorter form is derived through proteolytic processing or alternative splicing. Supporting the former possibility, BIS was shown to be cleaved upon apoptotic stresses mainly by caspase-3, resulting in the generation of a truncated product with an apparent mass of 40-kDa [14]. A subsequent study revealed that the caspase-mediated cleavage of BIS is a prerequisite for ubiquitination and degradation, linking to the loss of its pro-survival activity [15]. On the other hand, a band 10-kDa smaller than the expected mass was detected as the main band with an anti-BIS antibody in several tumor cell lysates, including LNCap and HCT116 cells [16]. However, neither the identification of any spliced form of BIS mRNA nor the physiological significance of the product has been reported yet.

In the present study, we demonstrated for the first time the existence of alternatively spliced BIS transcripts in which exon 3 or exons 2–3 are skipped. Furthermore, we also showed that SRSF3 is critically required for the inclusion of exon 3 of the BIS gene. Our findings suggest that the processing of the BIS pre-mRNA with coordination of the splicing factors is an important regulatory step for BIS expression and could be a potential therapeutic target.

## 2. Materials and Methods

### 2.1. Cell Culture and Transfection

293T, HCT116, and 8988T cells were maintained in DMEM (Biowest, Nuaillé, France), and A549, LNCap, and A172 cells were maintained in RPMI 1640 (Biowest), supplemented with 10% (*v*/*v*) fetal bovine serum (FBS, Biowest). The mouse embryonic fibroblasts (MEF) were prepared as previously described [17]. The transfection of the plasmids and siRNAs was carried out using Fugene 6 (Promega, Madison, WI, USA) and G-fectin (Genolution, Seoul, Korea) or Lipofectamine 2000 (Invitrogen, Carlsbad, CA, USA) according to the manufacturers’ protocols. The siRNA sequences and concentrations are listed in Appendix A. 293T and A549 cells were mainly used for the biochemical analysis in the present study.

### 2.2. Reverse Transcription PCR (RT-PCR) and Quantitative Real-Time PCR (qRT-PCR)

Total RNA was isolated with TRIzol reagent (Invitrogen), and cDNA was synthesized from 1 μg of total RNA with PrimeScript RT Master Mix (Takara, Shiga, Japan). Then, RT-PCR was performed using the Accupower Gold Multiplex PCR PreMix (Bioneer, Daejeon, Korea) and the indicated primers. PCR products were loaded onto 1% agarose gels and visualized by staining with Dyne Loading Star (Dynebio, Seongnam, Korea). The sequences of each band were determined by the Sanger method after cloning into a pJET1.2/blunt Cloning Vector (Thermo Scientific, Waltham, MA, USA). If necessary, the intensity of each band was quantified using ImageJ software (version 1.48, National Institute of Health; NIH, Bethesda, MD, USA). qRT-PCR was performed using the SYBR premix Ex Taq (Takara), with specific primers on the CFX96 Connect TM Real-Time PCR Detection System (Bio-Rad, Hercules, CA, USA). The relative values for the target mRNAs were calculated after normalization to the β-actin control. The specific primers for each mRNA are listed in Appendix A.

### 2.3. Western Blot Assay and Subcellular Fractionation

Cell protein extracts were prepared, and a Western blotting analysis was carried out as previously described, following standard procedures [18,19]. Briefly, whole-cell lysates were prepared in RIPA buffer (50 mM Tris, 150 mM NaCl, 1% NP-40, 0.5% sodium deoxycholate, 0.1% sodium dodecyl sulfate, and pH 7.5) with a protease inhibitor cocktail (Roche, Basel, Switzerland) and phosphate inhibitor (Sigma-Aldrich, St. Louis, MO, USA). For nuclear and cytosolic fractionation, cells were lysed with an RGB buffer (10 mM Tris-HCl pH 7.4, 10 mM NaCl, and 3 mM MgCl_2_), containing digitonin (40 μg/mL) and incubated on ice for 10 min. After centrifugation at 4 °C and 2100× *g* for 8 min, the supernatant was used as the cytosolic fraction. The pellet was further centrifuged two times with RGB buffer containing digitonin at the same conditions, lysed in RIPA buffer, and then used as the nuclear fraction. After protein quantification with a bicinchoninic acid (BCA) assay (Thermo Fisher Scientific), equal amounts of proteins were separated on SDS-PAGE gels and transferred to nitrocellulose membranes. After incubation in 5% skim milk for 30 min, the membranes were incubated overnight against specific antibodies: rabbit polyclonal anti-BIS serum (1:10000) [2], mouse monoclonal anti-β-actin (1:5000; Sigma-Aldrich), mouse monoclonal anti-SRSF3 (1:1000; Santa Cruz Biotechnology, Dallas, TX, USA), rabbit polyclonal anti-HSF1 (1:1000; Enzo Life Science, Farmingdale, NY, USA), mouse monoclonal anti-HSP70 (1:1000; Enzo Life Science), goat polyclonal anti-Lamin B (1:1000; Santa Cruz Biotechnology), or mouse monoclonal anti-GAPDH (1:1000; Santa Cruz Biotechnology). Next, the membranes were incubated with the appropriate secondary antibodies: anti-mouse IgG-HRP (1:5000; GeneTex, Irvine, CA, USA), anti-rabbit IgG-HRP (1:5000; GeneTex), and anti-goat IgG-HRP (1:1000; Thermo Fisher Scientific). The immunoreactive proteins were visualized using an enhanced chemiluminescence system (ECL Western Blotting Substrate; Promega).

### 2.4. Construction of Expression Vectors

The expression vectors encoding human BIS wild type (BIS-WT), human truncated BIS depleted in exon 3 (BIS-ΔE3), or exons 2–3 (BIS-ΔE2E3) were prepared by PCR amplification with primers including start and stop codons, as well as with cDNA from SRSF3-knockdown 293T cells, and subsequently inserted into a pcDNA3.1(-) vector (Addgene, Watertown, MA, USA). For the GFP-BIS wild type (WT) minigene reporter construct, the PCR products of human BIS genomic DNA, including exons 2–3 and part of the introns, were ligated into the pEGFP-C1 vector (Addgene), and exon 4 and part of the intron were ligated into pJET1.2/blunt (Thermo Fisher Scientific). Next, the first fragment with the GFP coding sequence and second fragment were ligated into the pcDNA3.1(-) vector. To generate the mutant minigene in which the SRSF3 binding site is deleted (GFP-BIS-Mut), we performed overlapping PCR reactions. A simple map for the minigene and the primers used for cloning are provided in Appendix A and Appendix A.

### 2.5. Biotin Pull-Down Assay

For in vitro transcription, DNA templates containing the T7 binding sequence were prepared by RT-PCR or direct synthesis (Bionics). The sequences of the DNA template are listed in Appendix A. Biotin (Bio-11-CTP; Enzo Life Science)-labeled RNAs were synthesized from DNA templates with the Maxiscript T7 kit (Invitrogen) and purified using the NucAway spin column (Thermo Fisher Scientific). Subsequently, 200 ng of each biotin-labeled RNA probe was incubated with 500 μg of whole-cell lysate from A549 or 293T cells overnight at 4 °C, followed by incubation with 30 μL dynabeads M-280 streptavidin (Invitrogen) for 4 h at 4 °C to capture the biotin-labeled RNAs selectively. The bead complexes were washed three times with PBS and boiled with 1× sample buffer (12 mM Tris-HCl pH 6.8, 25% glycerol, 2% sodium dodecyl sulfate, 5% 2-mercaptoethanol, and 0.1% bromophenol blue) at 95 °C for 5 min. Samples were analyzed by Western blot assay with an anti-SRSF3 antibody (Cruz Biotechnology). As a negative control, both the streptavidin-only control and the biotin-RNA control included in the Maxiscript T7 kit were used. The SRSF3 potential binding sites of BIS mRNA were predicted using the SpliceAid 2 program (http://193.206.120.249/splicing_tissue.html).

### 2.6. Statistics

The data are presented as the mean ± standard error of the mean (SEM). Statistical significance between two groups or multiple groups was analyzed by Student’s *t*-test or one-way ANOVA and Newman–Keuls multiple comparison test. A *p*-value of ≤0.05 was considered statistically significant.

## 3. Results

### 3.1. Identification of BIS mRNA Isoforms in Human Cell Lines and Mouse Tissues

To examine if the isoform of BIS mRNA exists, we performed RT-PCR using the primer sets covering exon 1 and exon 4 of human and mouse BIS mRNA, respectively (Figure 1A). Agarose electrophoresis of PCR products revealed that, in addition to BIS wild type (WT) (band I), two shorter products of approximately 600 bp and 250 bp (band II and band III) were observed consistently in several human cell lines and mouse tissues such as liver, lung, skeletal muscle, or heart (Figure 1B). DNA sequencing results indicated that the two smaller products, band II and band III, were BIS isoforms without exon 3 or exons 2–3, respectively (hereafter referred to as BIS-ΔE3 and BIS-ΔE2E3, respectively) (Figure 1C). Thus, these results proved the presence of isoforms of BIS mRNA in human and mouse cells, although in a small amount.

### 3.2. SRSF3 Is Involved in the Inclusion of Exon 3 of BIS Pre-mRNA

The two key families of splicing factors are serine/arginine-rich (SR) proteins and heterogeneous nuclear ribonucleoproteins (hnRNPs) [20]. To examine which splicing factor is involved in the alternative splicing of BIS pre-mRNA, the expression levels of these representative splicing factor families as well as HuR, SF3B2, and SF3B4 were suppressed using specific siRNA in A549 cells. Of the splicing factors tested, only SRSF3 knockdown reduced the BIS-WT mRNA level and increased the two smaller forms of BIS mRNA, band II and band III, respectively (Figure 2A). The effect of SRSF3 knockdown on the splicing of BIS pre-mRNA was also observed in other cell lines, such as 293T, HCT116, 8988T, or A172 cells (Figure 2B). The smaller products accumulated after SRSF3 knockdown were subsequently identified as BIS-ΔE3 and BIS-ΔE2E3, respectively (Figure 2C). A time-dependent profile following SRSF3 knockdown showed the skipping of exon 3 as early as 24 h, while the skipping of exons 2–3 was observed at 48 h at a lower intensity (Figure 2D). It was of note that, as SRSF3 decreased, a slightly smaller band than WT increased gradually (marked with an asterisk). However, this band was found to be an artifact generated when two templates of different lengths were amplified with the same primers (Appendix A). Next, to investigate whether these isoforms can be expressed as truncated proteins, we conducted a Western blot assay for BIS proteins after knockdown of SRSF3. To compare the exact molecular weight, the constructs for BIS-WT, -ΔE3, or -ΔE2E3 were transfected into 293T cells. As shown in Figure 2E, the knockdown of SRSF3 produced a truncated BIS protein, which showed an equivalent molecular weight with BIS-ΔE3 expressed by the exogenous construct. However, the BIS-ΔE2E3 protein was not detected upon SRSF3 knockdown, indicating that an increase in the BIS-ΔE2E3 on RT-PCR following SRSF3 depletion might be due to the increase in the availability as a template because of its shorter length than BIS-ΔE3 and BIS-WT. Collectively, depletion of SRSF3 promotes exon 3 skipping in BIS pre-mRNA, resulting in the stable production of truncated BIS protein.

The involvement of SRSF3 in the inclusion of exon 3 of BIS pre-mRNA was also verified using a mini-reporter gene in which BIS exon 2–4 was included, contiguous with a GFP sequence in a pcDNA3.1(-) vector (GFP-BIS; Figure 3A). The GFP-BIS minigene was expressed in 293T cells, followed by knockdown of SRSF3. Figure 3B shows that knockdown of SRSF3 induced a significant increase in exon 3 skipping in GFP-BIS to 15.6-fold, although to a lesser degree in endogenous BIS, which showed a 50-fold increase. Taken together, SRSF3 is essential for the inclusion of exon 3 of BIS pre-mRNA, which is reproducible in an exogenous GFP-BIS minigene.

### 3.3. Interaction of SRSF3 with Exon 3 Is Essential for the Inclusion of Exon 3 of BIS Pre-mRNA

Splicing factors are known to modulate skipping or inclusion of alternative exons by recognizing distinct RNA sequences [21,22]. To investigate if the interaction of SRSF3 with BIS pre-mRNA is required for SRSF3-mediated inclusion of exon 3 of BIS pre-mRNA, we prepared four types of biotin-labeled RNA oligomers derived from the four exons of BIS mRNA, and performed an RNA binding analysis using an in vitro pull-down assay. As shown in Figure 4A, SRSF3 binds to biotin-labeled exon 3 with strong affinity and exon 4 with weak affinity. Considering that exon 4 is twice as long as exon 3, we tentatively concluded that SRSF3 preferentially binds to exon 3 of BIS mRNA. We predicted the potential SRSF3 binding sites in exon 3 of BIS mRNA based on public program and arbitrarily divided exon 3 into four sections of 80 bp (a, b, c, and d) to include the potential binding sites (Figure 4B, upper). When comparing the binding capacity of each RNA probe from exon 3 to SRSF3, the RNA probe E3a in exon 3 was shown to interact strongly with SRSF3 (Figure 4B lower). We noticed that the potential binding motif for SRSF3, UCAUC, was repeated twice (UCAUCCUCAUC) in E3a. As a next step, we deleted the 11 nt of repeated SRSF3 binding motifs in probe E3a (E3a-Mut) and examined its binding activity to SRSF3. As shown in Figure 4C, the deletion of 11 nt potential binding motifs resulted in a significant reduction in the binding affinity with SRSF3 compared with E3a-wild-type (E3a-WT), both in 293T cells and A549 cells. Thus, these findings indicate that SRSF3 specifically interacts with the binding motifs in exon 3 close to the 3′ splice sites. To verify whether the interaction of SRSF3 with exon 3 of BIS mRNA is responsible for the inclusion of exon 3 of BIS mRNA, we generated the GFP-BIS minigene mutant in which 11 nt of potential binding sequences were deleted (Mut minigene). When expressed in A549 cells, the inclusion of exon 3 in the Mut minigene was profoundly decreased: skipping exon 3 in the Mut minigene was significantly increased to about 10-fold compared with the GFP-BIS minigene wild type (WT minigene; Figure 4D, lanes 1 and 3), which was comparable to the SRSF3 knockdown effect on the WT minigene (Figure 4D, lanes 2 and 3). The loss of binding site for SRSF3 in exon 3 promoted exon 3 skipping of BIS pre-mRNA; thus, this binding site can be considered as a strong exonic splicing enhancer (ESE) region. Figure 4D also shows that the skipping of exon 3 in the Mut minigene was further accelerated by SRSF3 depletion, indicating that another SRSF3 binding site or a mechanism might contribute to the inclusion of exon 3 of BIS pre-mRNA. Collectively, our results suggest that interaction of SRSF3 with an ESE in exon 3 is important to assure the intact linear BIS mRNA containing all exons.

### 3.4. Depletion of SRSF3 Inhibited HSF1 Translocation by Heat Shock Stress

In order to define the physiological significance of BIS-ΔE3, we examined the changes in the expression pattern of mRNA and protein of BIS during various cellular processes. However, we did not observe significant increases in the spliced product upon induction of apoptosis, autophagy, or senescence in 293T, A549 cells, or mouse embryonic fibroblasts [17,19] (Appendix A). BIS is a transcriptional target of HSF1 and also has been shown to interact with HSF1 in proteomic analyses [7,8,9,23]. Recently, we demonstrated that BIS is phosphorylated upon oxidative stress via ERK activation and that the deletion of phosphorylation sites in the protein stretch encoded by exon 3 attenuated the translocation of HSF1 upon oxidative stress [18]. Moreover, the splicing factor SF3B1 was shown to affect the activation of HSF1 via the mRNA level [24]. Therefore, we investigated whether SRSF3 modulates HSF1 activity alone or in cooperation with BIS. As shown in Figure 5A, knockdown of SRSF3 suppressed induction of the HSP70 protein, which was attributable to the decrease in HSP70 mRNA upon heat shock stress, both in A549 and 293T cells. Next, we tested the possibility that SRSF3 affects HSF1 translocation. Figure 5B shows that knockdown of SRSF3 resulted in the decrease of the HSF1 level in the nuclear fraction, while accumulating in the cytosol, indicating that SRSF3 affects HSF1 activity via modulation of the translocation step. Considering that the depletion of SRSF3 produced truncated BIS protein in which the region for affecting HSF1 translocation is deleted, it is possible that truncated BIS protein will impair the translocation of HSF1 in SRSF3-depleted cells. To test this possibility, 293T cells were co-transfected with SRSF3 siRNA and BIS-WT or BIS-ΔE3 expression vector and exposed to heat shock. As shown in Figure 5C, the HSF1 levels in the nuclear fraction were restored by overexpression of BIS-WT but not by BIS-ΔE3, which was inversely correlated with the HSF1 levels in the cytosolic fraction. In addition, the alteration of translocation of HSF1 was directly linked to HSP70 induction, as determined by Western blotting and qRT-PCR (Figure 5D). Therefore, our findings indicated that maintenance of the SRSF3 levels is important for heat shock response upon various cellular stresses, probably via the maintenance of the integrity of BIS pre-mRNA containing exon 3, leading to an intact BIS protein, which is important for shutting down HSF1.

## 4. Discussion

In the present study, using PCR and sequencing in various cell lines, we identified for the first time the presence of alternatively spliced forms of BIS mRNA in which exon 3 or exons 2–3 are skipped. We also demonstrated that among the 22 splicing factors tested, knockdown of SRSF3 alone resulted in the skipping of exon 3 of BIS pre-mRNA, indicating the critical role for SRSF3 in the inclusion of exon 3 of BIS pre-mRNA. SR proteins have been known to control the inclusion or skipping of specific exons through interacting with one specific recognition sequence in the target alternative exons [21,25]. Thus, we investigated whether SRSF3 functions by directly binding to the potential binding site. Using an in vitro pull-down assay, we observed that SRSF3 most strongly binds to exon 3 of BIS pre-mRNA. Subsequently, we predicted several potential binding sites for SRSF3 on exon 3 using the SpliceAid 2 program. A putative binding site, composed of a repeated sequence of UCAUC (UCAUCCUCAUC) close to the 3′-splice site on exon 3, was shown to be an SRSF3 responsive element based on the following evidence. First, among four RNA probes from exon 3, the RNA probe containing the repeated sequences strongly interacted with SRSF3. Secondly, the deletion of 11 bp of the repeated sequence in the 80 bp of RNA probe from exon 3 attenuated the binding affinity with SRSF3. Thirdly, the deletion of the putative binding site in the GFP-BIS minigene resulted in the skipping of exon 3, even in the presence of SRSF3, indicating that the putative binding site for SRSF3 is one major ESE region of exon 3 (Figure 4A–D). Thus, SRSF3 promotes exon 3 inclusion through interaction with an ESE region in the target exon of BIS pre-mRNA. However, it should be noted that additional regulatory elements including enhancers and silencers might be involved for the determination of skipping or inclusion of exon3 of BIS pre-mRNA in vivo.

SRSF3, the smallest member of the SR protein family, has known involvement in the alternative splicing of pre-mRNA of numerous genes, including *TP53*, *CD44*, *FOXM1*, *INSR*, and *KLF6*, as well as its own mRNA, affecting various cellular functions such as proliferation, motility, and metabolism [25,26,27,28,29,30,31,32,33,34]. Our results clearly show that BIS is a new target of SRSF3, and BIS mRNA deleted in exon 3 was stably translated into protein. However, we did not find the acceleration of the skipping of exon 3 of BIS mRNA accompanied increases in the truncated protein levels upon induction of several processes that are associated with the function of BIS, such as apoptosis, autophagy, and senescence (Appendix A). In addition, MEF exhibits no difference in the pattern for inclusion or skipping of exon 3 compared with A549 cells, excluding the requirement of truncated BIS protein during embryogenesis and function (Appendix A). Therefore, SRSF3 might be required for the constitutive splicing of BIS pre-mRNA to ensure the inclusion of exon 3 in the correct order, by preventing exon skipping by interaction with exon 3 of the BIS mRNA.

Interestingly, SRSF3 is included in the list of proteins that bind BIS, as detected by affinity purification followed by mass spectrometry (AP-MS) [35]. Furthermore, overexpression of BIS promotes its own expression, namely autoregulation, through activation of HSF1 and subsequent activation of its own promoter harboring heat shock response elements [36]. Thus, to test the possibility that BIS is involved in its own splicing process, we examined whether SRSF3-mediated regulation of exon 3 inclusion occurs in BIS knockout A549 cells [37]. We found that exon 3 is skipped by SRSF3 knockdown in the absence of BIS protein (data not shown). These findings imply that BIS–SRSF3 interaction or the BIS protein itself is not required for SRSF3-mediated inclusion of exon 3 in BIS pre-mRNA.

A previous report demonstrated the short truncated BIS protein in LNCap and HCT116 human cells via a Western blot assay [16]; but, in our results, the agarose electrophoresis patterns of the PCR products from those cells were not different from those from the A549 and 293T cells (Figure 1B). Furthermore, the Western blot assay also revealed no difference in the molecular weight of the main band of the BIS protein in LNCap and HCT116 cells compared with the other two cells (Appendix A). Thus, the difference in the molecular weight of BIS in these cells between previous studies and our study might be due to the protein extraction procedure rather than a difference in the spliced pattern of BIS pre-mRNA.

Recently, we identified the induction of BIS phosphorylation at Thr285 and Ser289 in the protein stretch encoded by exon 3 under oxidative stress occurred in an ERK-dependent manner, which was important for releasing HSF1 into the nucleus [18]. However, heat shock stress did not induce phosphorylation of the BIS protein even though BIS was also shown to be involved in the shuttling of HSF1 under heat stress [38]. Thus, under heat shock stress, BIS might regulate the localization of HSF1 by a mechanism other than phosphorylation. On the other hand, SRSF3 was shown to be spliced into the truncated form by stressful conditions [39,40]. Based on these studies, we examined the possible roles of SRSF3 and BIS in the nuclear translocation of HSF1 under heat shock stress. Unlike oxidative stress, splicing of SRSF3 into truncated SRSF3 protein was not induced by heat shock stress in A549 or 293T cells (data not shown). However, SRSF3 depletion significantly attenuated the induction of HSP70 mRNA, which was attributable to the reduced translocation of HSF1 into the nucleus (Figure 5A,B). Subsequently, we showed that the reduction of HSF1 translocation by SRSF3 depletion was restored by overexpression of BIS-WT, but not by BIS deleted in exon 3, indicating that the SRSF3 depletion-mediated reduction of HSF1 translocation was partly due to the increase in the spliced form of BIS. Our results thus provide a novel role for SRSF3 in the modulation of HSF1 localization, requiring an intact BIS protein, including exon 3. However, the molecular mechanism by which SRSF3 and BIS perform nuclear translocation of HSF1 is to be determined by further studies.

The association of splicing factor and activation of HSF1 was demonstrated by a previous study, showing that knockdown or pharmacological inhibition of SF3B1 resulted in a significant decrease in heat shock response via downregulation of HSF1 transcription. In addition, aberrant mutations of SF3B1 in a gain-of-function form and high expression of SRSF3 are commonly associated with some types of cancers [24,34]. In this context, it can be hypothesized that overexpression of SRSF3 might confer resistance to cancer cells upon various stresses through the promotion of nuclear location of HSF1 in cooperation with BIS. HSF1 activation, in turn, might increase the transcription of BIS and HSP70 harboring HSE in the promoter, which exerts anti-apoptotic and protein quality control functions, providing a more pro-survival milieu for cancer cells. However, we did not show the overexpression effect of SRSF3 on the inclusion of exon 3 of BIS mRNA and the subsequent effect on HSF1 translocation, which could be regarded as a limitation of the present study. As one possible reason, the amount of BIS mRNA skipped in exon 3 was so small that it seems difficult to quantitatively measure the change. Even though the amount of the spliced form of BIS pre-mRNA is detected by RT-PCR, their actual quantity should be very small because of the amplification power of a PCR.

Since the inclusion of exon 3 of BIS mRNA is critically determined by SRSF3 levels, the maintenance of SRSF3 at the appropriate physiological levels is indispensable for the synthesis of intact BIS protein. Therefore, the change in BIS mRNA in linearity and subsequent outcome on cellular fate should be investigated for the conditions in which SRSF3 is reduced, or, alternatively, the conditions in which the spliced form of SRSF3 is increased, such as exposure to alcohol or oxidative stress [39,40,41].

In summary, we demonstrate that SRSF3 is a critical determinant for the inclusion of exon 3 BIS pre-mRNA. The relevance of SRSF3 and BIS in various pathophysiological conditions should be investigated in further studies.

## Figures and Tables

**Figure 1 cells-09-02325-f001:**
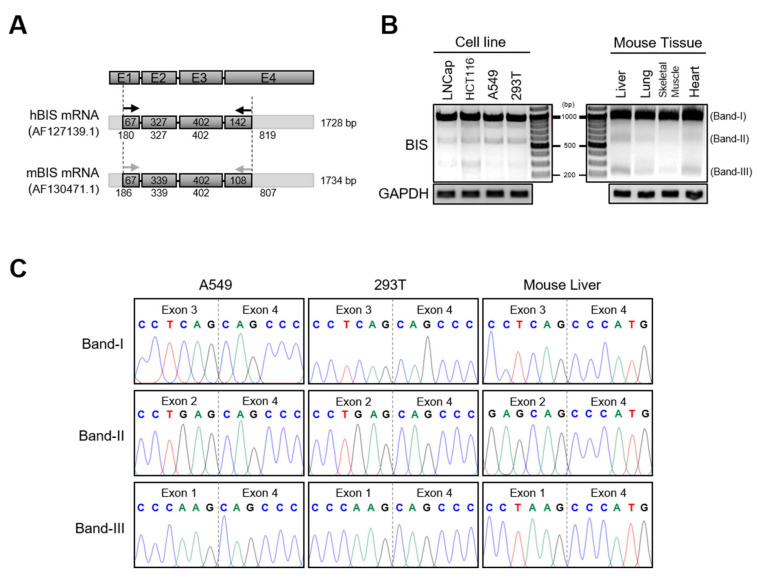
The identification of skipping of exon 2 and/or exon 3 of BIS mRNA in human cells and mouse tissues. (**A**) Schematic diagram of four exons (E1–E4) in human and mouse BIS mRNA. The positions of the primer sets for PCR are marked with arrows. (**B**) Reverse transcription was performed with mRNA from the indicated sources, and subsequent PCR products of BIS transcripts were analyzed by 1% agarose electrophoresis. The assumed isoforms of BIS were indicated as band I, II, and III. (**C**) Sanger sequencing results for band I, II, and III in (**B**) indicated that band II and III lacked exon 3 and exons 2–3, respectively, compared with full-length BIS mRNA (band I).

**Figure 2 cells-09-02325-f002:**
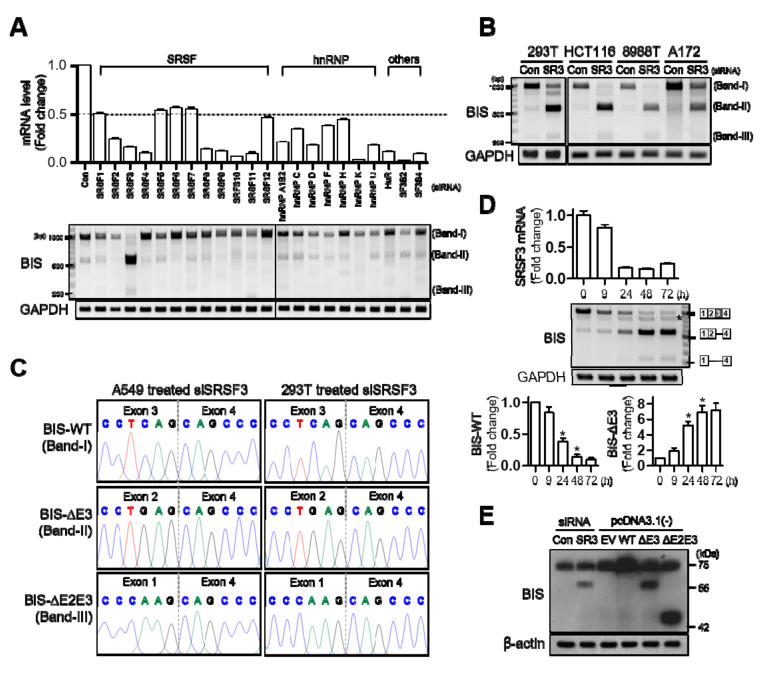
SRSF3 knockdown promotes exon 3 skipping of endogenous BIS pre-mRNA in various cell lines. (**A**) Expression of representative splicing factors, including the SRSF3 family, hnRNP family, and several other factors, was determined by qRT-PCR following treatment with specific siRNA in A549 cells (upper graph). RT-PCR for BIS transcripts was performed using primer sets in Figure 1A (lower panel). C, control siRNA. Note that knockdown of SRSF3 resulted in an increase of band II and III. (**B**) BIS transcripts were analyzed by RT-PCR using RNA extracted from 293T, HCT116, 8988T, and A172 cells after SRSF3 knockdown were treated with 100 nM control siRNA (Con) or SRSF3 siRNA (SR3) for 48 h. (**C**) Band II and III accumulated after the SRSF-depletion was verified as BIS-ΔE3 and BIS-ΔE2E3, respectively, by DNA sequencing in A549 and 293T cells. (**D**) Time-dependent profile of BIS mRNA splicing after SRSF3 knockdown in A549 cells. SRSF3 levels were determined by qRT-PCR at the indicated times after treatment with 50 nM SRSF3 siRNA (upper). The alternative splicing of BIS was analyzed by agarose electrophoresis at the indicated times after SRSF3 knockdown (middle). Schematic diagram for RNA products is presented on the right. The additional band marked with an asterisk is an artifact of RT-PCR. Refer to Appendix A for details. Quantification of BIS-WT and -ΔE3 PCR product levels were determined from three independent experiments (lower). * *p* < 0.05. (**E**) 293T cells were transfected with 100 nM of the control (Con) or SRSF3 siRNA (SR3), and then the BIS proteins were analyzed by Western blot assay. Exogenous expression of BIS was also analyzed after transfection with 1 μg empty vector (EV), BIS-WT, BIS-∆E3, or BIS-∆E2E3 in a pcDNA 3.1(-) vector for 24 h, respectively. Beta-actin was used as the loading control.

**Figure 3 cells-09-02325-f003:**
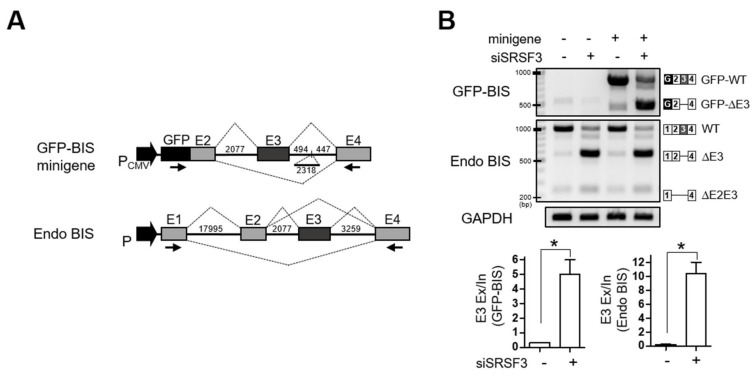
SRSF3 depletion accelerates exon 3 skipping in the GFP-BIS minigene. (**A**) Schematic diagrams of the GFP-BIS minigene and human endogenous BIS (Endo BIS) gene showing the lengths of each exon and intron between the indicated exons. Broken lines above or below the introns show the splicing direction. In the GFP-BIS minigene, 2318 bp were deleted in the intron between exon 3 and 4. The locations of primer sets for RT-PCR are marked with arrows. (**B**) The effect of SRSF3 depletion on the GFP-BIS minigene was analyzed with RT-PCR. 293T cells were transfected with 1 μg of GFP-BIS minigene together with 100 nM of control siRNA or SRSF3 siRNA for 24 h (upper). RNA products are schematically presented on the right. Quantification of RNA products from lanes 3 and 4 are presented as exon 3 (E3) exclusion/inclusion (Ex/In) ratios of GFP-BIS and endogenous (Endo) BIS on the lower bar graphs. Values shown are the mean ± SEM of three independent experiments. * *p* < 0.05.

**Figure 4 cells-09-02325-f004:**
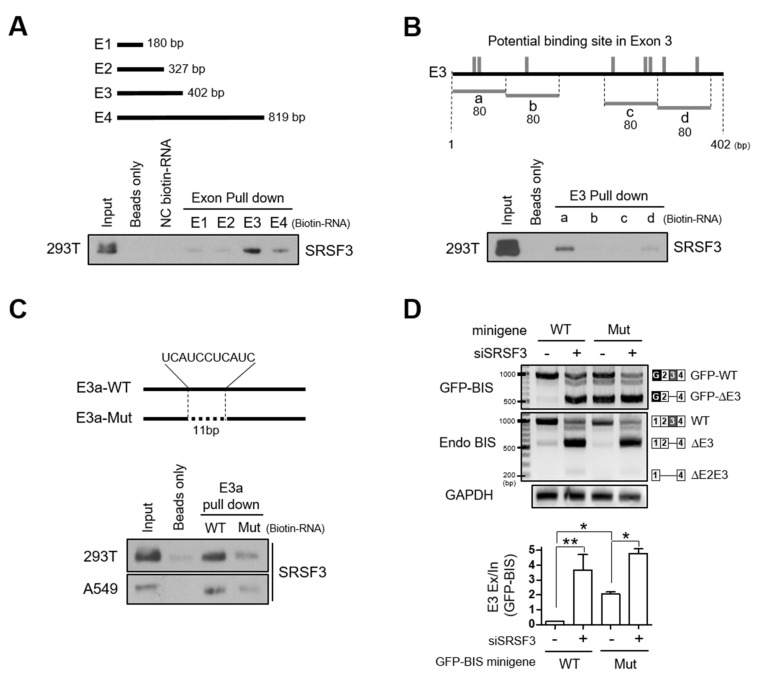
SRSF3-dependent inclusion of exon 3 in BIS pre-mRNA is mediated by interaction with exon 3. (**A**) Schematic diagrams of four exons of BIS showing the corresponding length of each exon (upper). The biotin-labeled RNA probes of each BIS exon were pulled down with streptavidin beads with 293T cell lysates and analyzed by Western blotting using anti-SRSF3 antibody (lower). Beads and negative control (NC) biotin-RNA provided by the manufacturer were used as negative controls, according to the manufacturer’s protocol. (**B**) Schematic mapping for SRSF3 potential binding sites on BIS exon 3 mRNA (vertical gray lines, upper). Four types of RNA probes with 80 bp (a, b, c, and d) derived from exon 3 (E3) were labeled with biotin and subsequently processed for binding ability to SRSF3 by biotin pull-down assays. (**C**) The 11 bp putative binding site for SRSF3 in E3a-wild-type (E3a-WT) was deleted in the deletion mutant (E3a-Mut, upper). The interaction with two probes with SRSF3 was examined in 293T or A549 cells via biotin pull-down assays. (**D**) A549 cells were transfected with 1 μg GFP-BIS minigene wild type (WT minigene) or GFP-BIS minigene mutant, in which the 11 bp potential binding sites were deleted (Mut minigene), followed by transfection of 100 nM of SRSF3 siRNA. RT-PCR analysis for GFP-BIS and endogenous (Endo) BIS transcripts is shown in the upper panels. Quantification of exon 3 skipping in the WT or Mut minigene was presented as Exon 3 (E3) exclusion/inclusion (Ex/In) ratios. Data are the mean ± SEM of three independent experiments. The statistical significance was determined using ANOVA and Newman–Keuls multiple comparison test. * *p* < 0.05, ** *p* < 0.01.

**Figure 5 cells-09-02325-f005:**
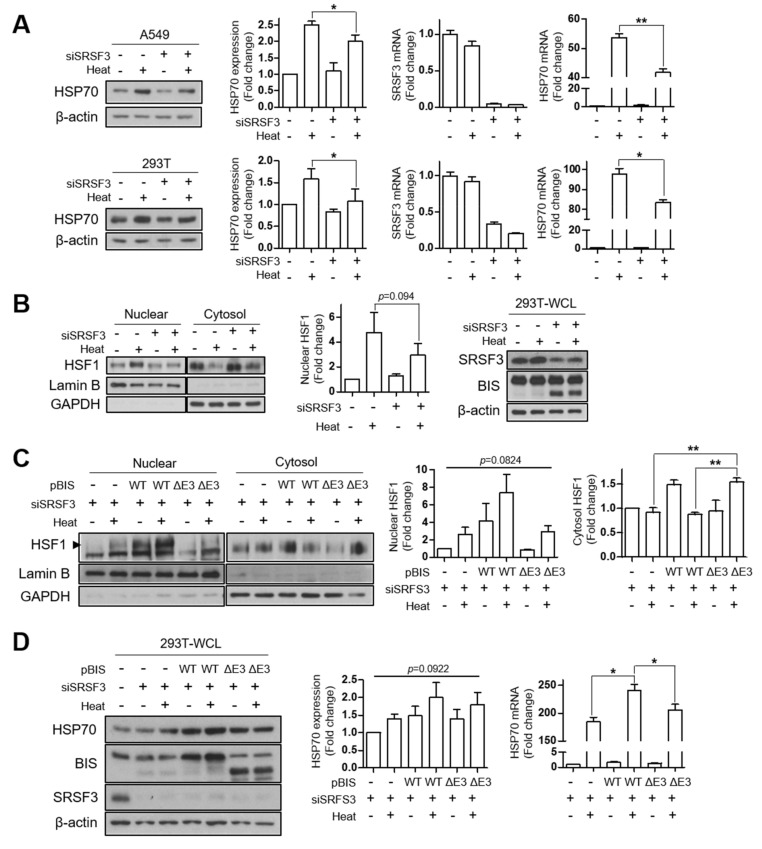
The effect of SRSF3 depletion on HSF1 translocation upon heat shock stress. (**A**) A549 (upper) and 293T (lower) cells were transfected with 100 nM of SRSF3 siRNA (siSRSF3) for 24 h followed by heat shock stress at 43 °C or 42 °C on the water bath, respectively, for 30 min and then incubated at 37 °C for another 3 h for recovery. HSP70 induction was examined by Western blot analysis (left panels) and quantified using ImageJ software (second panels). The relative SRSF3 and HSP70 mRNA levels were determined by qRT-PCR after normalization with those of β-actin (third and fourth panels). (**B**) Western blotting for HSF1 levels in the nuclear/cytosol fractions in 293T cells under the same conditions as in (**A**) (left panel). Lamin B and GAPDH levels were used as the loading controls for nuclear and cytosol fraction markers, respectively. The nuclear HSF1 protein levels compared to those in the control cells were determined from three independent experiments (middle panel). The Western blotting results for SRSF3, BIS, and β-actin in 293T whole-cell lysate under the same conditions (right panel). (**C**) The HSF1 levels were determined in the nuclear/cytosol fractions of 293T cells transfected with 100 nM of SRSF3 siRNA for 24 h and subsequently with 1 μg of BIS-WT or BIS-ΔE3 plasmid for 24 h, and then exposed to heat shock stress. Nuclear accumulation of HSF1 was increased by BIS-WT, not by BIS-ΔE3, as shown in the representative Western blot results (left panel) and by the quantification graph of the HSF1 protein levels in the nuclear or cytosol fractions (middle and right panels), determined from three independent experiments. (**D**) Expression levels of HSP70 were determined at protein levels and mRNA levels under the same conditions as (**C**). Western blot analysis was performed for HSP70, BIS, and SRSF3 expression (left panel), and the quantification of HSP70 protein levels, as well as mRNA levels, are presented in the middle and right panels, respectively, from three independent experiments. For the determination of statistical significance, we performed Student’s *t*-test (**A**,**B**) or ANOVA and Newman–Keuls multiple comparison test (**C**,**D**). * *p* < 0.05, ** *p* < 0.01.

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
