# Peer review of "SRSF3 Is a Critical Requirement for Inclusion of Exon 3 of BIS Pre-mRNA"

_cells, 2020, doi:10.3390/cells9102325_

Round 1

Reviewer 1 Report

The Authors in their work, discovered for the first time the existence of alternatively spliced BIS transcripts with skipped exon 3 or exon 2-3 and regulation of this process by SRSF3. The topic of the manuscript is attractive, and this work is well written. Overall, the experiments were well designed and carried out. However, some points need clarification:

  • The dilutions of antibodies used in this work should be given.
  • The Reviewer could not evaluate the results from chapter 3.2. due to the lack of figure 2.
  • The section: results contain fragments that should be included in the discussion section, for example, lines 147-154.
  • There is an incorrect legend of Figure 5A.
  • Minor grammar and spelling mistakes.

Author Response

The Authors in their work, discovered for the first time the existence of alternatively spliced BIS transcripts with skipped exon 3 or exon 2-3 and regulation of this process by SRSF3. The topic of the manuscript is attractive, and this work is well written. Overall, the experiments were well designed and carried out. However, some points need clarification:

: We appreciate your encouraging and positive comments with kindness.

The dilutions of antibodies used in this work should be given.

: We added the dilution information of antibodies in the Materials and Method section as follows.

 After incubation in 5% skim milk for 30 min, the membranes were incubated overnight against specific antibodies: rabbit polyclonal anti-BIS serum (1:10000) [2], mouse monoclonal anti-êžµ-actin (1:5000; Sigma-Aldrich), mouse monoclonal anti-SRSF3 (1:1000; Santa Cruz Biotechnology, Dallas, TX, USA), rabbit polyclonal anti-HSF1 (1:1000; Enzo Life Science, Farmingdale, NY, USA), mouse monoclonal anti-HSP70 (1:1000; Enzo Life Science), goat polyclonal anti-Lamin B (1:1000; Santa Cruz Biotechnology), or mouse monoclonal anti-GAPDH (1:1000; Santa Cruz Biotechnology). Next, the membranes were incubated with the appropriate secondary antibodies: anti-mouse IgG-HRP (1:5000; GeneTex, Irvine, CA, USA), anti-rabbit IgG-HRP (1:5000; GeneTex), and anti-goat IgG-HRP (1:1000; Thermo Fisher Scientific). The immunoreactive proteins were visualized using an enhanced chemiluminescence system (ECL Western Blotting Substrate; Promega). (Page 3, Lines 98-107)

The Reviewer could not evaluate the results from chapter 3.2.

: We apologize for this inconvenience. We have rearranged the figures and the text in this section to make it more easier to read.

The section: results contain fragments that should be included in the discussion section, for example, lines 147-154.

: According to the reviewer’s suggestion, we moved the descriptions to the Discussion section as follows.

A previous report demonstrated the short truncated BIS protein in LNCap and HCT116 human cells via western blot assay [16]; but, in our results, the agarose electrophoresis patterns of PCR products from those cells were not different from those from A549 and 293T cells (Figure 1B). Furthermore, western blot assay also revealed no difference in the molecular weight of the main band of BIS protein in LNCap and HCT116 cells compared with the other two cells (Figure S4). Thus, the difference in the molecular weight of BIS in these cells between previous studies and our study might be due to the protein extraction procedure rather than a difference in the spliced pattern of BIS pre-mRNA. (Page 10, Lines 358-365)

There is an incorrect legend of Figure 5A.

: We sincerely apologize for our careless mistake. We corrected the mark for heat shock as follows (see attachment).

Minor grammar and spelling mistakes.

: We rechecked the entire manuscript for English and have made appropriate revisions.

Reviewer 2 Report

This paper reports the identification of novel splice isoforms of BIS and the regulation of exon 3 inclusion by SRSF3.  The data supporting these aspects of the paper is strong and the results are sound.  The manuscript is a little weaker when it comes to a functional outcome of the altered splicing.  The functional data is that knockdown of SRSF3 slightly reduces the induction of HSP70, and reduces the nuclear translocation of HSF. The authors admit that no changes in BIS splicing occur with various insults (Fig. S4).  So what is the physiological relevance of this axis?

The HSF localization results are somewhat puzzling as SRSF3 knockdown only increases the ex3- variant slightly, the majority of BIS is still full length.  Using 293T cells they show that transfection of WT BIS increases the impaired nuclear translation of HSF1 but exon3- doesn't (Fig. 5C).  We don't know if this is truly a rescue as the control cells without SRSF3 kd are not shown.   In any case why does SRSF3 kd impair nuclear localization of HSF when most BIS is still full length?  Does the ex3- variant have a dominant negative effect?  What does ex3- BIS do when transfected in to normal cells?  The model is that BIS interacts with HSF through exon 3 and that BIS is required for nuclear localization of HSF.  So the ex3- should not interact with HSF (need to confirm that) but still could interfere if other domains are need to the localization.  Can the amount of nuclear and cytoplasmic HSF1 be quantified absolutely?  Do they add a standard to each gel so between gel companions can be made?  How much HSF1 is nuclear vs cytoplasmic?  Can this be confirmed by staining?

Another concern is the quantification of the isoforms.  Densitometry on stained RT-PCR agarose gel is not accurate.  The intensity of staining is roughly proportional to the mass of DNA fragment as the de can intercalate randomly.  So a band twice the size will have twice the intensity for the same molar amount.  This method of quantification is thus only good when the sizes of the two isoforms are very similar.  The other issue, which the authors discuss, is that the amplification efficiency can be very different for PCR amplicons that are very different in size.  The authors invoke this to explain the lack of BISex3- protein while the amount of mRNA is fairly substantial.  So they are saying that the RT-PCR is a great overestimate of the amount.  

There is also the possibility that translation of the ex3- protein may be reduced.  The exon is in frame so the resulting mRNA should not be subject to NMD, but it might be worth checking.  Have the authors verified that the ex3- mRNA is intact at the 3'end, or does it have any other alterations?

Minor points:

Size markers are shown in Fig1B but are then missing from subsequent figures.

Line 292, I think it should read but not by BIS-E3.  Missing the not.

Fig 5C. what bands are we looking at for HSF1? The upper or lower?

line 320 various not variable

line 322 among the 22 splicing factors tested, not kinds of splicing factors

line 351-2. Needs rewriting - BIS auto regulates its own expression.  How does BIS autoregluate?

line 386 Over expression of a SRSF3 cDNA is not subject to the same regulation as the endogenous gene.  It is easy to overexposes the protein.  Some viral promoters are sensitive to SRSF3 - (eg Adenoviral promoters) so that may effect expression.

Reviewer 3 Report

This review is for the manuscript entitled “SRSF3 is a critical requirement for inclusion of exon 3 of BIS pre-mRNA” authored by Jeong-Hwa Lee and colleagues.

Results: Depletion of SRSF3 promoted skipping of exon 3 of BIS pre-mRNA. Furthermore, authors demonstrate that SRSF3 specifically interacts with the putative binding sites in exon 3, in which deletion promoted skipping of exon 3 in GFP-BIS minigene. SRSF3 depletion, accompanied by production of truncated BIS protein, inhibited the nuclear translocation ofHSF1, which was restored by full-length but not exon 3-deleted BIS.

Conclusions: The authors conclude that SRSF3 is a critical determinant for inclusion of exon 3 BIS pre-mRNA and downstream heat shock signaling.

Review: Overall, this article is well-researched and the data are generally of high quality. Furthermore, these findings are an important contribution to defining the alternative splicing program in cancer. However, there are a few major/moderate concerns which are detailed below:

Major criticism 1: Figure 4: Although this reviewer considers it highly likely that SRSF3 is interacting with the CAUCC putative SRSF3 consensus sequence within exon 3 of BIS based on the authors’ findings, a binding assay demonstrating direct interaction would greatly increase enthusiasm for this manuscript.

Moderate criticism 1: In the statistics section, the authors indicate that they have only used the student’s t-test for these studies. This may not be sufficient for the multiple sample assessments shown in figures 4 and 5. Appropriate statistical methods for multiple means testing would be beneficial to this manuscript.

Author Response

Review: Overall, this article is well-researched and the data are generally of high quality. Furthermore, these findings are an important contribution to defining the alternative splicing program in cancer. However, there are a few major/moderate concerns which are detailed below:

  • Major criticism 1: Figure 4: Although this reviewer considers it highly likely that SRSF3 is interacting with the CAUCC putative SRSF3 consensus sequence within exon 3 of BIS based on the authors’ findings, a binding assay demonstrating direct interaction would greatly increase enthusiasm for this manuscript.

:  We appreciate your comment. As suggested, the direct interaction between SSF3 and the putative binding site for SRSF3 in the exon 3 of BIS mRNA should be demonstrated to prove our hypothesis. However, it was difficult to perform these experiments within the time allocated for the revision (10 days). So please understand that we cannot show the direct interaction between SRSF3 and its putative target in exon 3 of BIS mRNA at this time. Even though it is not directly binding, the in vitro pull-down assay with RNA oligomer with total cell lysates would support our presumption substantially.

  • Moderate criticism 1: In the statistics section, the authors indicate that they have only used the student’s t-test for these studies. This may not be sufficient for the multiple sample assessments shown in figures 4 and 5. Appropriate statistical methods for multiple means testing would be beneficial to this manuscript.

: We appreciate the valuable suggestion. We performed one-way ANOVA and Newman-Keuls multiple comparison analysis for Figure 4D and Figure 5C and D. When the p-value is not within the significant range, we present the value in the graph. When the values are significant, the p-values are presented as asterisks as follows.

Reviewer 4 Report

In general, the manuscript shows interesting and novel findings regarding alternative splicing regulation of BIS. The manuscript presents rational and sound results.

I only found minor spelling and grammar mistakes that should be corrected before acceptance.

I suggest to edit the title for figure 1 because it seems too descriptive and should indicate the conclusion of the results instead.

Author Response

In general, the manuscript shows interesting and novel findings regarding alternative splicing regulation of BIS. The manuscript presents rational and sound results. I only found minor spelling and grammar mistakes that should be corrected before acceptance. I suggest to edit the title for figure 1 because it seems too descriptive and should indicate the conclusion of the results instead.

: We appreciate your encouraging and positive comments with kindness.

As suggested, we changed the title for Figure 1 as follows.

Figure. 1. The identification of skipping of exon 2 and/or exon 3 of BIS mRNA in human cells and mouse tissues.

And we re-checked the spelling and grammar mistakes in the manuscript and made appropriate revisions.

Round 2

Reviewer 3 Report

The authors have adequately answered the criticisms. Although inclusion of evidence of direct binding would be nice, this reviewer agrees that the timeline for those studies is not feasible.

Author Response

We appreciate for your encouraging comments.